# Context-sensitive active sensing in humans

**Sheeraz Ahmad**
Department of Computer Science and Engineering
University of California San Diego
9500 Gilman Drive La Jolla, CA 92093
sahmad@cs.ucsd.edu

**He Huang**
Department of Cognitive Science
University of California San Diego
9500 Gilman Drive La Jolla, CA 92093
heh001@ucsd.edu

**Angela J. Yu**
Department of Cognitive Science
University of California San Diego
9500 Gilman Drive La Jolla, CA 92093
ajyu@ucsd.edu

## Abstract

Humans and animals readily utilize active sensing, or the use of self-motion, to focus sensory and cognitive resources on the behaviorally most relevant stimuli and events in the environment. Understanding the computational basis of natural active sensing is important both for advancing brain sciences and for developing more powerful artificial systems. Recently, we proposed a goal-directed, context-sensitive, Bayesian control strategy for active sensing, C-DAC (Context-Dependent Active Controller) (Ahmad & Yu, 2013). In contrast to previously proposed algorithms for human active vision, which tend to optimize abstract statistical objectives and therefore cannot adapt to changing behavioral context or task goals, C-DAC directly minimizes behavioral costs and thus, automatically adapts itself to different task conditions. However, C-DAC is limited as a model of human active sensing, given its computational/representational requirements, especially for more complex, real-world situations. Here, we propose a myopic approximation to C-DAC, which also takes behavioral costs into account, but achieves a significant reduction in complexity by looking only one step ahead. We also present data from a human active visual search experiment, and compare the performance of the various models against human behavior. We find that C-DAC and its myopic variant both achieve better fit to human data than Infomax (Butko & Movellan, 2010), which maximizes expected cumulative future information gain. In summary, this work provides novel experimental results that differentiate theoretical models for human active sensing, as well as a novel active sensing algorithm that retains the context-sensitivity of the optimal controller while achieving significant computational savings.

## 1 Introduction

Both artificial and natural sensing systems face the challenge of making sense out of a continuous stream of noisy sensory inputs. One critical tool the brain has at its disposal is *active sensing*, a goal-directed, context-sensitive control strategy that prioritizes sensing and processing resources toward the most rewarding or informative aspects of the environment (Yarbus, 1967). Having a formal understanding of active sensing is not only important for advancing neuroscientific progress but also developing context-sensitive, interactive artificial agents.

The most well-studied aspect of human active sensing is saccadic eye movements. Early work suggested that saccades are attracted to *salient* targets that differ from surround in one or more of feature dimensions (Koch & Ullman, 1985; Itti & Koch, 2000); however, saliency has been found to only account for a small fraction of human saccadic eye movement (Itti, 2005). More recently, models of human active vision have incorporated top-down objectives, such as maximizing the expected future cumulative informational gain (Infomax) (Lee & Yu, 2000; Itti & Baldi, 2006; Butko & Movellan, 2010), and maximizing the one-step look-ahead probability of finding the target (greedy MAP)(Najemnik & Geisler, 2005). However, these are generic statistical objectives that do not naturally adapt to behavioral context, such as changes in the relative cost of speed versus error, or the energetic or temporal cost associated with switching from one sensing location/configuration to another. We recently proposed the C-DAC (Context-Dependent Active Controller) algorithm (Ahmad & Yu, 2013), which maps from Bayesian posterior beliefs about the environment into the action space while optimizing directly with respect to context-sensitive, behavioral goals; C-DAC was shown to result in better accuracy and lower search time, as compared to Infomax and greedy MAP, in various simulated task environments.

In this paper, we investigate whether human behavior is better explained by taking into account task-specific considerations, as in C-DAC, or whether it is sufficient to optimize a generic goal, like that of Infomax. We compare C-DAC and Infomax performance to human data, in terms of fixation choice and duration, from a visual search experiment. We exclude greedy MAP from this comparison, based on the results from our recent work showing that it is an almost random, and thus highly suboptimal strategy for the well-structured visual search task presented here.

At a theoretical level, both Infomax and C-DAC are offline algorithms involving iterative computation until convergence, and which compute a global policy that specifies the optimal action (relative to their respective objectives) for every possible setting of previous actions and observations, most of which may not be used often or at all. Both of these algorithms suffer the well-known curse of dimensionality, and are thus difficult, if not impossible, to generalize to more complex, real-world problems. Humans seem capable of planning and decision-making in very high-dimensional settings, while readily adapting to different behavioral context. It therefore behooves us to find a computationally inexpensive strategy that is nevertheless context-sensitive. Here, we consider an approximate algorithm that chooses actions online and myopically, by considering the behavioral cost of looking only one step ahead (instead of an infinite horizon as in the optimal C-DAC policy).

In Sec. 2, we briefly summarize C-DAC and Infomax, as well as introduce the myopic approximation to C-DAC. In Sec. 3, we describe the experiment, present the human behavioral data, and compare the performance of different models to the human data. In Sec. 4, we simulate scenarios where C-DAC and myopic C-DAC achieve a flexible trade-off between speed, accuracy and effort depending on the task demands, whereas Infomax falls short – this forms experimentally testable predictions for future investigations. We conclude in Sec. 5 with a discussion of the insights gained from both the experiment and the models, as well as directions for future work.

## 2   The Models

In the following, we assume a basic active sensing scenario, which formally translates to a sequential decision making process based on noisy inputs, where the observer can control both the sampling location and duration. For example, in a visual search task, the observer controls where to look, when to switch to a different sensing location, and when to stop searching and report the answer. Although the framework discussed below applies to a broad range of active sensing problems, we will use language specific to visual search for concreteness.

### 2.1   C-DAC

This model consists of both an inference strategy and a control/decision strategy. For inference, we assume the observer starts with a prior belief over the latent variable (true target location), and then updates her beliefs via Bayes rule upon receiving each new observation. The observer maintains a probability distribution over the $k$ possible target locations, representing the corresponding belief about the presence of the target in that location (belief state). Thus, if $s$ is the target location (latent), $\boldsymbol{\lambda}_t := \{\lambda_1, \ldots, \lambda_t\}$ is the sequence of fixation locations up to time $t$ (known), and

$\mathbf{x}_t := \{x_1, \ldots, x_t\}$ is the sequence of observations up to time $t$ (observed), the belief state and the belief update rule are:

$$\mathbf{p}_t := (P(s = 1 | \mathbf{x}_t; \boldsymbol{\lambda}_t), \ldots, P(s = k | \mathbf{x}_t; \boldsymbol{\lambda}_t))$$

$$\mathbf{p}_t^i = P(s = i | \mathbf{x}_t; \boldsymbol{\lambda}_t) \propto p(x_t | s = i; \lambda_t) P(s = i | \mathbf{x}_{t-1}; \boldsymbol{\lambda}_{t-1}) = f_{s,\lambda_t}(x_t) \mathbf{p}_{t-1}^i \tag{1}$$

where $f_{s,\lambda}(x_t)$ is the likelihood function, and $\mathbf{p}_0$ the prior belief distribution over target location.

For the decision component, C-DAC optimizes the mapping from the belief state to the action space (continue, switch to one of the other sensing locations, stop and report the target location) with respect to a behavioral cost function. If the target is at location $s$, and the observer declares it to be at location $\delta$, after spending $\tau$ units of time and making $n_\tau$ number of switches between potential target locations, then the total cost incurred is given by:

$$l(\tau, \delta; \boldsymbol{\lambda}_\tau, s) = c\tau + c_s n_\tau + \mathbf{1}_{\{\delta \neq s\}} \tag{2}$$

where $c$ is the cost per unit time, $c_s$ is the cost per switch, and cost of making a wrong response is 1 (since we can always make one of the costs to be unity via normalization). For any given policy $\pi$ (mapping belief state to action), the expected cost is $L_\pi := c\mathbb{E}[\tau] + c_s\mathbb{E}[n_s] + P(\delta \neq s)$. At any time $t$, the observer can either choose to stop and declare one of the locations to be the target, or choose to continue and look at location $\lambda_{t+1}$. Thus, the expected cost associated with stopping and declaring location $i$ to be the target is:

$$\bar{Q}_t^i(\mathbf{p}_t, \boldsymbol{\lambda}_t) := \mathbb{E}[l(t, i) | \mathbf{p}_t, \boldsymbol{\lambda}_t] = ct + c_s n_t + (1 - \mathbf{p}_t^i) \tag{3}$$

And the minimum expected cost for continuing sensing at location $j$ is:

$$Q_t^j(\mathbf{p}_t = \mathbf{p}, \boldsymbol{\lambda}_t) := c(t+1) + c_s(n_t + \mathbf{1}_{\{j \neq \lambda_t\}}) + \min_{\tau', \delta, \boldsymbol{\lambda}_{\tau'}} \mathbb{E}[l(\tau', \delta) | \mathbf{p}_0 = \mathbf{p}, \lambda_1 = j] \tag{4}$$

The value function $V(\mathbf{p}, i)$, or the expected cost incurred following the optimal policy ($\pi^*$), starting with the prior belief $\mathbf{p}_0 = \mathbf{p}$ and initial observation location $\lambda_1 = i$, is:

$$V(\mathbf{p}, i) := \min_{\tau, \delta, \boldsymbol{\lambda}_\tau} \mathbb{E}[l(\tau, \delta) | \mathbf{p}_0 = \mathbf{p}, \lambda_1 = i] . \tag{5}$$

Then the value function satisfies the following recursive relation (Bellman, 1952), and the action that minimizes the right hand side is the optimal action $\pi^*(\mathbf{p}, k)$:

$$V(\mathbf{p}, k) = \min\left( \left( \min_i \bar{Q}_1^i(\mathbf{p}, k) \right), \min_j \left( c + c_s \mathbf{1}_{\{j \neq k\}} + \mathbb{E}[V(\mathbf{p}', j)] \right) \right) \tag{6}$$

This can be solved using dynamic programming, or more specifically value iteration, whereby we guess an initial value of the value function and iterate eq. 6 until convergence.

## 2.2 Infomax policy

Infomax (Butko & Movellan, 2010) presents a similar formulation in terms of belief state representation and Bayesian inference, however, for the control part, the goal is to maximize long term information gain (or minimize cumulative future entropy of the posterior belief state). Thus, the action-values, value function, and the resultant policy are:

$$Q^{im}(\mathbf{p}_t, j) = \sum_{t'=t+1}^{T} \mathbb{E}[H(\mathbf{p}_{t'}) | \lambda_{t+1} = j]; \quad V^{im}(\mathbf{p}_t, j) = \min_j Q^{im}(\mathbf{p}_t, j); \quad \lambda_{t+1}^{im} = \operatorname*{argmin}_j Q^{im}(\mathbf{p}_t, j)$$

Infomax does not directly prescribe when to stop, since there are only continuation actions and no *stopping action*. A general heuristic used for such strategies is to stop when the confidence in one of the locations being the target (the belief about that location) exceeds a certain threshold, which is a

free parameter challenging to set for any specific problem. In our recent work we used an optimistic strategy for comparing Infomax with C-DAC by giving Infomax a stopping boundary that is fit to the one computed by C-DAC. Here we present a novel theoretical result that gives an inner bound of the stopping region, obviating the need to do a manual fit. The bound is sensitive to the sampling cost $c$ and the signal-to-noise ratio of the sensory input, and underestimates the size of the stopping region.

Assuming that the observations are binary and Bernoulli distributed (i.i.d. conditioned on target and fixation locations), i.e.:

$$f_{s,\lambda}(x) = p(x|s=i; \lambda=j) = \mathbf{1}_{\{i=j\}}\beta^x(1-\beta)^{1-x} + \mathbf{1}_{\{i\neq j\}}(1-\beta)^x\beta^{1-x} \tag{7}$$

We can state the following result:

**Theorem 1.** *If $p^*$ is the solution of the equation:*

$$p\frac{(2\beta-1)(1-p)}{\beta p + (1-\beta)(1-p)} = c$$

*where $c$ is the cost per unit time as defined in sec. 2.1, then for all $\mathbf{p}^i > p^*$, the optimal action is to stop and declare location $i$ under the cost formulation of C-DAC.*

*Proof.* The cost incurred for collecting each new sample is $c$. Therefore stopping is optimal when the improvement in belief from collecting another sample is less than the cost incurred to collect that sample. Formally, stopping and choosing $i$ is optimal for the corresponding belief $\mathbf{p}^i = p$ when:

$$\max_{p'\in\mathcal{P}}(p') - p \leq c$$

where $\mathcal{P}$ is the set of achievable beliefs starting from $p$. Furthermore, if we solve the above equation for equality, to find $p^*$, then by problem construction, it is always optimal to stop for $p > p^*$ (stopping cost $(1-p) < (1-p^*)$). Given the likelihood function $f_{s,\lambda}(x)$ (eq. 7), we can use eq. 1 to simplify the above relation to:

$$p\frac{(2\beta-1)(1-p)}{\beta p + (1-\beta)(1-p)} = c$$

$\square$

## 2.3 Myopic C-DAC

This approximation attempts to optimize the contextual cost proposed in C-DAC, but only for one step in the future. In other words, the planning is based on the inherent assumption that the next action is the last action permissible, and so the goal is to minimize the cost incurred in this single step. The actions thus available are, stop and declare the current location as the target, or choose another sensing location before stopping. Similar to eq. 6, we can write the value function as:

$$V(\mathbf{p}, k) = \min\left((1-\mathbf{p}^k), \min_j\left(c + c_s\mathbf{1}_{\{j\neq k\}} + \min_{l_j}\left(1 - \mathbb{E}[\mathbf{p}^{l_j}]\right)\right)\right) \tag{8}$$

where $j$ indexes the possible sensing locations, and $l_j$ indexes the possible stopping actions for the sensing location $j$.

Note that the value function computation does not involve any recursion, just a comparison between simple-to-compute action values for different actions. For the visual search problem considered below, because the stopping action is restricted to only the current sensing location, $l_j = j$, the right-hand side simplifies to

$$V(\mathbf{p}, k) = \min\left((1-\mathbf{p}^k), \min_j\left(c + c_s\mathbf{1}_{\{j\neq k\}} + \left(1 - \mathbb{E}[\mathbf{p}^j]\right)\right)\right)$$

$$= \min\left((1-\mathbf{p}^k), \min_j\left(c + c_s\mathbf{1}_{\{j\neq k\}} + \left(1 - \mathbf{p}^j\right)\right)\right) \tag{9}$$

the last equality due to $\mathbf{p}$ being a martingale. It can be seen, therefore, that this myopic policy overestimates the size of the stopping region: if there is only step left, it is never optimal to continue looking at the same location, since such an action would not lead to any improvement in expected accuracy, but incur a unit cost of time $c$. Therefore, in the simulations below, just like for Infomax, we set the stopping boundary for myopic C-DAC using the bound presented in Theorem 1.

# 3 Case Study: Visual Search

In this section, we apply the different active sensing models discussed above to a simple visual search task, and compare their performance with the observed human behavior in terms of accuracy and fixation duration.

## 3.1 Visual search experiment

The task involves finding a target (the patch with dots moving to the left) amongst two distractors (the patches with dots moving to the right), where a patch is a stimulus location possibly containing the target. The definition of target versus distractor is counter-balanced across subjects. Fig. 1 shows schematic illustration of the task at three time points in a trial. The display is gaze contingent, such that only the location currently fixated is visible on the screen, allowing exact measurement of where a subject obtains sensory input. At any time, the subject can declare the current fixation location to be the target by pressing space bar. Target location for each trial is drawn independently from the fixed underlying distribution $(1/13, 3/13, 9/13)$, with the spatial configuration fixed during a block and counter-balanced across blocks. As search behavior only systematically differed depending on the probability of a patch containing a target, and not on its actual location, we average data across all configurations of spatial statistics and differentiate the patches only by their prior likelihood of containing the target; we call them patch 1, patch 3, and patch 9, respectively. The study had 11 participants, each presented with 6 blocks (counterbalanced for different likelihoods: $3! = 6$), with each block consisting of 90 trials, leading to a total of 5940 trials. Subjects were rewarded points based on their performance, more if they got the answer correct (less if they got it wrong), and penalized for total search time as well as the number of switches in sensing location.

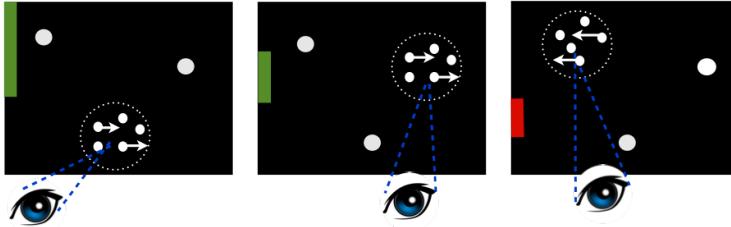

Figure 1: Simple visual search task, with gaze contingent display.

## 3.2 Comparison of Model Predictions and Behavioral Data

In the model, we assume binary observations (eq. 7), which are more likely to be 1 if the location contains the target, and more likely to be 0 if it contains a distractor (the probabilities sum to 1, since the left and right-moving stimuli are statistically/perceptually symmetric). We assume that within a block of trials, subjects learn about the spatial distribution of target location in that block by inverting a Bayesian hidden Markov model, related to the Dynamic Belief Model (DBM) (Yu & Cohen, 2009). This implies that the target location on each trial is generated from a categorical distribution, whose underlying rates at the three locations are, with probability $\alpha$, the same as last trial and, probability $1 - \alpha$, redrawn from a prior Dirichlet distribution. Even though the target distribution is fixed in a block, we use DBM with $\alpha = 0.8$ to capture the general tendency of human subjects to typically rely more on recent observations than distant ones in anticipating upcoming stimuli. We assume that subjects choose the first fixation location on each trial as the option with the highest prior probability of containing the target. The subsequent fixation decisions are made following a given control policy (C-DAC, Infomax or Myopic C-DAC).

We investigate how well these policies explain the emergence of a certain confirmation bias in humans – the tendency to favor the more likely (privileged) location when making a decision about target location. We focus on this particular aspect of behavioral data because of two reasons: (1) The more obvious aspects (e.g. where each policy would choose to fixate first) are also the more trivial ones that all reasonable policies would display (e.g. the most probable one); (2) Confirmation

bias is a well studied, psychologically important phenomenon exhibited by humans in a variety of choice and decision behavior (see (Nickerson, 1998), for a review), and is, therefore, important to capture in its own right.

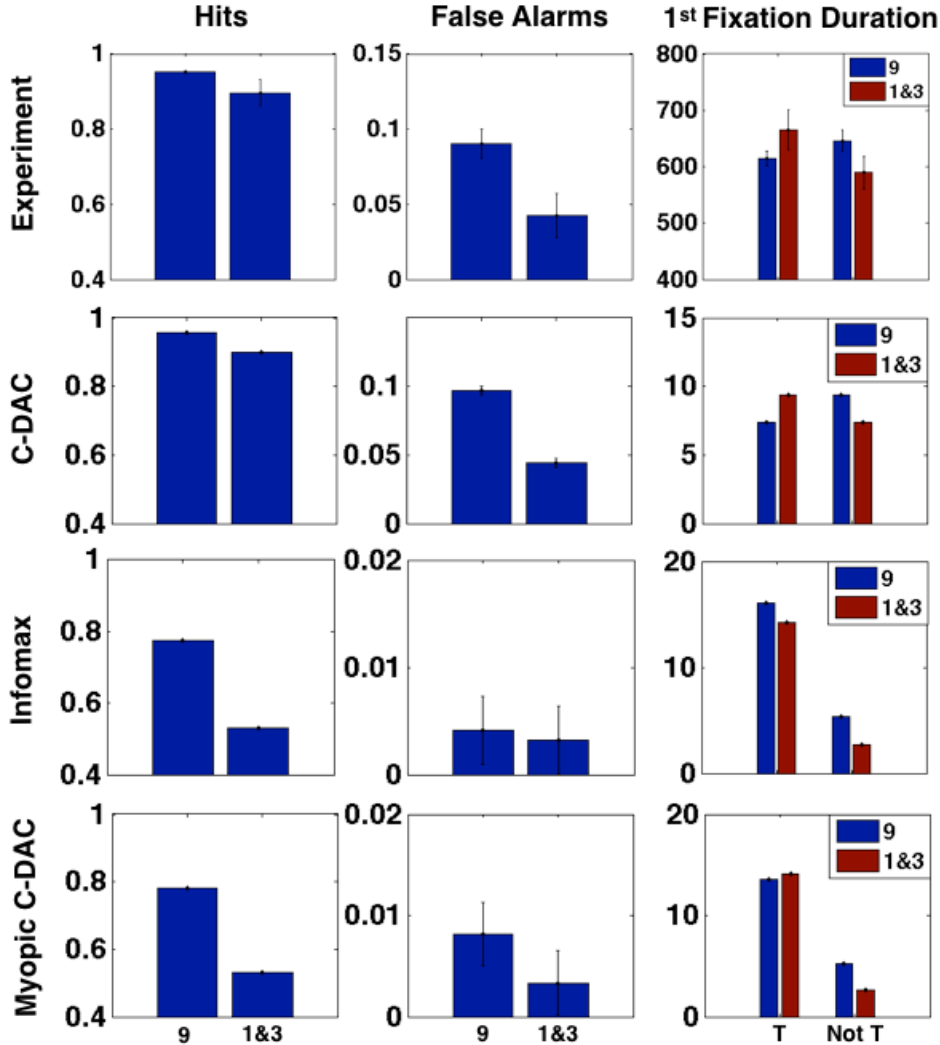

Figure 2: Confirmation bias in human data and model simulations. The parameters used for C-DAC policy are $(c, c_s, \beta) = (0.005, 0.1, 0.68)$. The stopping thresholds for both Infomax and myopic C-DAC are set using the bound developed in Theorem 1. The spatial prior for each trial, used by all three algorithms, is produced by running DBM on the actual experimental stimulus sequences experienced by subjects. Units for fixation duration: millisecond (experiment), number of time-steps (simulations)

Based on the experimental data (Fig. 2), we observe this bias in fixation choice and duration. Subjects are more likely to identify the 9 patch to contain the target, whether it is really there ("hits", left column) or not ("false alarms", middle column). This is not due to a potential motor bias (tendency to assume the first fixation location contains the target, combined with first fixating the 9 patch most often), as we only consider trials where the subject first fixates the relevant patch. The confirmation bias is also apparent in fixation duration (right column), as subjects fixate the 9 patch shorter than the 1 & 3 patches when it is the target (as though faster to confirm), and longer when it is not the target (as though slower to be dissuaded). Again, only those trials where the first fixation landed on the relevant patch are included. As shown in Figure 2, these confirmation bias phenomena are captured by both C-DAC and myopic C-DAC, but not by Infomax.

Our results show that human behavior is best modeled by a control strategy (C-DAC or myopic C-DAC) that takes into account behavior costs, e.g. related to time and switching. However, C-DAC in its original formulation is arguably not very psychologically plausible. This is because C-DAC requires using dynamic programming (recursing Bellman's optimal equation) offline to compute a *globally* optimal policy over the continuous state space (belief state), so that the discretized state space scales exponentially in the number of hypotheses. We have previously proposed families of parametric and non-parametric approximations, but these still involve large representations, and recursive solutions. On the other hand, myopic C-DAC incurs just a constant cost to compute the policy online for only the current belief state, is consequently psychologically more plausible, and provides a qualitative fit to the data with a simple threshold bound. We believe its performance can be improved by using a tighter bound to approximate the stopping region. Infomax, on the other hand, is not context sensitive, and our experiments suggest that even manually setting its threshold to match that of C-DAC does not lead to substantial improvement in performance (not shown).

## 4 Model Predictions

With the addition of the parametric threshold to Infomax and myopic C-DAC, we discover the wider disparity which we earlier observed between C-DAC and Infomax disappears for a large class of parameter settings, since now the stopping boundary for Infomax is also context sensitive. Similar claim holds for myopic C-DAC. However, one scenario where Infomax does not catch up to the full context sensitivity of C-DAC, is when cost of switching from one sensing location to another comes in to play. This is due to the rigid switching boundaries of Infomax. In contrast, myopic C-DAC can adjust its switching boundary depending on context. We illustrate the same for the case when $(c, c_s, \beta) = (0.1, 0.1, 0.9)$ in Fig. 3.

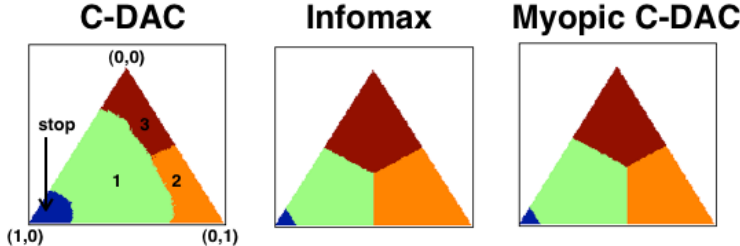

Figure 3: Different policies for the environment $(c, c_s, \beta) = (0.1, 0.1, 0.9)$, as defined on the belief state $(\mathbf{p}^1, \mathbf{p}^2)$, under affine transform to preserve rotational symmetry. Blue: stop & declare. Green: fixate location 1. Orange: fixate location 2. Brown: fixate location 3.

We show in Fig. 4 how the differences in policy space translate to behavioral differences in terms of accuracy, search time, number of switches, and total behavioral cost (eq. 2). As with the previous results, we set the threshold using the bound developed in Theorem 1. Note that, as expected, the performance of Infomax and Myopic C-DAC are closely matched on all measures for the case $c_s = 0$. The accuracy of C-DAC is poorer as compared to the other two, because the threshold used for the other policies is more conservative (thus stopping and declaration happens at higher confidence, leading to higher accuracy), but C-DAC takes less time to reach the decision. Looking at the overall behavioral costs, we can see that although C-DAC loses in accuracy, it makes up at other measures, leading to a comparable net cost. For the case when $c_s = 0.1$, we notice that the accuracy and search time are relatively unchanged for all the policies. However, C-DAC has a notable advantage in terms of number of switches, while the number of switches remain unchanged for Infomax. This case exemplifies the context-sensitivity of C-DAC and Myopic C-DAC, as they both reduce number of switches when switching becomes costly. When all these costs are combined we see that C-DAC incurs the minimum overall cost, followed by Myopic C-DAC, and Infomax incurs the highest cost due to its lack of flexibility for a changed context. Thus Myopic C-DAC, a very simple approximation to a computationally complex policy C-DAC, still retains context sensitivity, whereas Infomax with complexity comparable to C-DAC falls short.

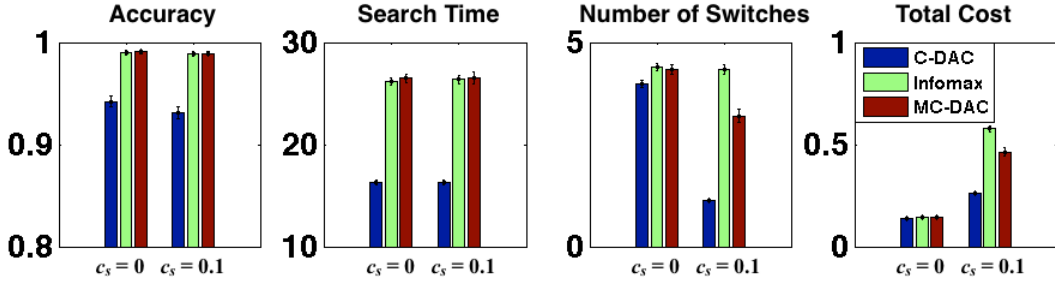

Figure 4: Comparison between C-DAC, Infomax and Myopic C-DAC (MC-DAC) for two environments $(c, c_s, \beta) = (0.005, 0, 0.68)$ and $(0.005, 0.1, 0.68)$. For $c_s > 0$, the performance of C-DAC is better than MC-DAC which in turn is better than Infomax.

## 5   Discussion

In this paper, we presented a novel visual search experiment that involves finding a target amongst a set of distractors differentiated only by the stimulus characteristics. We found that the fixation and choice behavior of subjects is modulated by top-down factors, specifically the likelihood of a particular location containing the target. This suggests that any purely bottom-up, saliency based model would be unable to fully explain human behavior. Subjects were found to exhibit a certain confirmation bias – the tendency to systematically favor a location that is a priori judged more likely to contain the target, compared to another location less likely to contain the target, even in the face of identical sensory input and motor state. We showed that C-DAC, a context-sensitive policy we recently introduced, can reproduce this bias. In contrast, a policy that aims to optimize statistical objectives of task demands and ignores behavioral constraints (e.g. cost of time and switch), such as Infomax (Lee & Yu, 2000; Itti & Baldi, 2006; Butko & Movellan, 2010), falls short. We proposed a bound on the stopping threshold that allows us to set the decision boundary for Infomax, by taking into account the time or sampling cost $c$, but that still does not sufficiently alleviate the context-insensitivity of Infomax. This is most likely due to both a sub-optimal incorporation of sampling cost and an intrinsic lack of sensitivity toward switching cost, because there is no natural way to compare a unit of switching cost with a unit of information gain. To set the stage for future experimental research, we also presented a set of predictions for scenarios where we expect the various models to differ the most.

While C-DAC does a good job of matching human behavior, at least based on the behavioral metrics considered here, we note that this does not necessarily imply that the brain implements C-DAC exactly. In particular, solving C-DAC exactly using dynamic programming requires a representational complexity that scales exponentially with the dimensionality of the search problem (i.e. the number of possible target locations), thus making it an impractical solution for more natural and complex problems faced daily by humans and animals. For this reason, we proposed a myopic approximation to C-DAC that scales linearly with search dimensionality, by eschewing a globally optimal solution that must be computed and maintained offline, in favor of an online, approximately and locally optimal solution. This myopic C-DAC algorithm, by retaining context-sensitivity, was found to nevertheless reproduce critical fixation choice and duration patterns, such as the confirmation bias, seen in human behavior. However, exact C-DAC was still better than myopic C-DAC at reproducing human data, leaving room for finding other approximations that explain brain computations even better. One possibility is to find better approximations to the switching and stopping boundary, since these together completely characterize any decision policy, and we previously showed that there might be a systematic, monotonic relationship between the decision boundaries and the different cost parameters (Ahmad & Yu, 2013). We proposed one such bound on the stopping boundary here, and other approximate bounds have been proposed for similar problems (Naghshvar & Javidi, 2012). Further investigations are needed to find more inexpensive, yet context-sensitive active sensing policies, that would not only provide a better explanation for brain computations, but yield better practical algorithms for active sensing in engineering applications.

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
