[Reviews · NeurIPS 2013]

Submitted by Assigned_Reviewer_1

This paper propose a myopic version of C-DAC, a visual search algorithm that includes costs and benefits. Some c-b algorithm of this sort will surely beat a naive cost-benefit free version...so the issue is just what the details look like.
Fig 2 shows that C-DAC does well on many dimensions of fit.

Comments:
C-DAC is not psycho'll plausible? why not?
Fig 2. why is set 9 and 1&3 so diagnostic.
Summary: This paper find evidence for a search algorithm that myopically adjust for the cost of future visual saccades.

Submitted by Assigned_Reviewer_4

The paper explores theoretical issues in active sensing -- situations where people much actively gather information to achieve some goal. A extension to a previous model (C-DAC) is proposed which myopically gathers information in order to maximize a task-specific cost function. The model is compared against an alternative that favors information without respect to cost (Infomax). The overall goal of the paper is interesting and important. The novelty of the paper is somewhat reduced simply because it reports an extension to Ahmad & Yu (2013) to deal myopically with action selection. This is fine but I wondered how novel or important that aspect of the modeling actually was (e.g., it didn't seem to figure centrally into explanations of the experiment itself).

There is some prior work on this issue. In particular, the authors might check out Markant & Gureckis, 2012 "Does the utility of information influence sampling behavior?" which explores a similar issue: whether people optimize information or some task-specific cost function. They found that information optimizing provided a better fit to people's behavior.

The paper was poorly written overall. The abstract was much too long, the paper exceeded the 8 page limit at NIPS, and the paper cites Ahmad & Yu, 2013 in almost every sentence including twice in the abstract alone! I had a really hard time understanding the experiment design. Terms like "patch" are introduced without definition (is a patch a set of trials with a particular distribution of target probabilities, or possible target locations? I couldn't really tell). Also, while the overall theoretical issue seemed well motivated, the paper itself focused on this "confirmatory search" phenomena which the authors seemed to ad-hoc identify in their data. Why was this phenomena selected or critical for differentiating the models?

Also, the model result for section 3 were not that compelling. The full C-DAC hits all the qualitative patterns as do the myopic one. The authors don't discuss what this means. Is it the case that the measures the authors selected here just don't tell any of the models apart very well? No fit statistics or quantification of the advantage for the more limited model is offered other than to describe is as "more psychologically plausible." Section 4 was also very opaque. I didn't understand the policy-space graphs in Figure 3 and I suspect that most readers will have significant difficulty with this as well. It is unclear what "Greedy MAP" is as well as this model is never mentioned in the paper.

The analysis presented in Figure 4 is odd because if this is meant to be a model of humans, no mention is made of human performance. It is fine to say that the myopic C-DAC model performs well on a number of metrics, but as a model it should match human observer as well (unless this is meant as a engineering analysis of a computer vision based visual search algorithm?).

*Note* My initial evaluation was a little more negative, but after discussion with the other reviewers I have given a 7. I think some of the critical comments above still apply, but there are also strengths including the emphasis on multiple model comparison, the empirical data, and the novelty of deriving theorems about the optimal stopping rule. I thought the author's response was helpful and it would be nice if the paper was edited somewhat in accordance with the reviewer questions to be more clear if it is accepted.
Summary: A myopic modification is added to a recent model of cost-sensitive active sensing. The model is fit to data from a visual search task, and the myopic model appears to fit the data of the experiment fairly well. However, qualitatively the full (non-myopic) model performs about the same. Other aspects of the paper were somewhat difficult to understand given what was written including the graphs of policy-space functions. Overall I think there are a lot of interesting ideas here, but the paper is not quite developed yet. The contribution beyond the original C-DAC model is unclear (outside of a vauge notion of being more psychologically plausible because it doesn't optimize multiple steps ahead into the future).

Submitted by Assigned_Reviewer_6

Paper 1349 – Context-sensitivity in human active sensing

This paper explores how people guide their search for information in a visual display. They describe a recently proposed model, C-DAC, present an optimal stopping result, and a computation-bounded approximation to the optimal cost. Afterwards, they present an experiment where there are three patches of stimuli (dots moving either to the left or right with some amount of coherence), and participants are supposed to report the patch with dots moving to the left as quickly as possible. Generally, the models using C-DAC generated policies that resulted in behavior a bit more like the human results, than the behavior resulting from a model using an Infomax generated policy. Finally, they conclude with simulations that can distinguish between the three policy generation methods.

The paper focuses on a critical computational problem faced by both people and machines: How should an agent allocate its sensors to the different places in the environment to best solve the task at hand? The approach taken is very interesting, elegant, and a reasonable candidate for what the visual system is aiming to solve at a computational level. However, the simulations at the end of the paper distinguishing between the different methods was very interesting, but I was left disappointed that there weren’t human results as well! Furthermore, I found the evaluation of their method too poor to merit a NIPS publication, and as, C-DAC is a previously published model, and it should not be considered a contribution of the paper.

However, there are some issues with the paper that leave me on the fence regarding whether or not I believe it should be published at NIPS. The framing of the paper is about how C-DAC provides context-sensitive policies, but I don’t understand in what sense it is context-sensitive. It would help to be more upfront about what exactly the context is in this context. I don’t see how the case study highlighted why being context-sensitive matters. In fact, besides Infomax getting the search times wrong for when 9 was the target, I did not find the experiment too convincing that people were guided by similar computational principles as C-DAC. Furthermore, there are few dependent variables being predicted for the complexity and number of parameters used by C-DAC. Finally, the simulations at the end of the paper seemed much more interesting than the case study and potentially could provide stronger evidence for C-DAC. However, there were no human experiments based on these simulations. The authors may be better off waiting to publish the project with the human results for these later simulations, which would result in a much stronger paper, than publishing the current paper.

Minor comments:
Section 2.1: When a vector has been indexed to a scalar, I personally like it to no longer be bold.
Section 3.1, line 232: “the probability of a target containing a target” should that be a patch containing a target?
Section 3.2 title: “Model vs. behavior” the use of vs. sounded a bit strange to me. Also, should model be plural?
Section 3.2, line 261-263: “… the general tendency for human subjects typically rely more on recent observations…” -> “… the general tendency for human subjects to rely more on recent observations…”
Figure 2: label the units for the y-axes.
Summary: An excellent project in the making; however, it seems too premature for publication.
Author Feedback

Author rebuttal: We thank the reviewers for their valuable feedback. We feel that the main concerns expressed by the reviewers may have risen from a lack of clarity in our writing in certain places, and the reviews were very helpful in highlighting these lapses. In the following, we first address the concerns shared by the reviewers, and then proceed to individual points as needed. We will of course also incorporate these additional clarifications in any revision of the paper.

General comments
1. The reviewers asked why C-DAC is "arguably not very psychologically plausible" (line 332). This is because C-DAC requires using dynamic programming (recursing Bellman's optimal equation) offline to compute a *globally* optimal policy over the continuous state space (belief state), so that the discretized state space scales exponentially in the number of hypotheses. Similar observations are made in the discussion section of Markant & Gureckis, 2012, the reference suggested by reviewer #2. Myopic C-DAC is "online, fast, and easy to scale to more complex problems" (line 412), as it incurs a constant cost to compute the policy online, for only the current belief state. It has huge computation savings and is thus "psychologically more plausible" (line 335).

2. The reviewers asked why we focused on the particular aspect of the behavioral data that we did. That is in part because the more obvious aspects (e.g. where each policy would choose to fixate first) are also the more trivial aspects that all reasonable policies would predict (e.g. the most probable one). We focused on the confirmation-bias-related fixation duration differences, because previous models like Infomax could not account for them. Furthermore, confirmation bias is also psychologically important. It is a well studied phenomenon exhibited by humans in a variety of choice and decision behavior (see Nickerson, 1998, for a review), and we feel any convincing model should be able to account for it.

3. The reviewers asked for a summary of how the performance of C-DAC was different from Myopic C-DAC and Infomax in capturing data. As shown in Fig. 2, out of the four confirmatory trends that follow, C-DAC captures all four, myopic C-DAC three (1, 2 and 4), and Infomax two (2 and 4): (1) Time taken to confirm when the more likely patch (9) is the target is shorter compared to when less likely patches (1&3) are the target (two left bars); (2) Similarly, time taken to disconfirm when patch 9 is not target is longer (two right bars); (3) Time taken to confirm patch 9 when it is the target is shorter than time taken to disconfirm when it is not (blue bars); (4) Time taken to confirm when patches 1&3 are targets is longer (red bars). Furthermore, C-DAC provides best quantitative fit for Hits and False Alarms.

4. The reviewers asked why we split the performance metrics into two sets (1&3 and 9). Since a major goal of the experiment is to examine top-down cognitive influences in saccadic vision, we manipulated top-down expectations by having the target appearing in the three patches with unequal probabilities, and looking at the fixation behavior in each patch separately. Note that the coherence of random dots and hence the bottom-up stimulus information is identical at each patch -- the stimulus being fixated is either always the target or distractor. So any differences in human or model behavior arise entirely from top-down expectations about target location. This is something that any bottom-up or saliency-based models cannot account for (e.g. Itti & Koch, 2000) .

5. The reviewers asked why we included model predictions that we do not yet have data for. We firmly believe that any good model should not only be able to explain existing data but also make specific and testable predictions for future experiments. Thus, Sec. 4 contains predictions that sets the stage for future investigations, highlighting scenarios where we predict the different models would agree or disagree the most. We believe that timely publication of these predictions would facilitate scientific progress by encouraging other groups to investigate these predictions either collaboratively or independently.


Individual Comments
Reviewer #2:
The reviewer inquired about our novel contributions. These are: (1) A novel visual search experiment with model comparisons to human behavior. (2) An efficient approximation, Myopic C-DAC, that compares favorably with human behavior. (3) A novel way to derive a decision threshold bound, that can be used to augment policies like Infomax, which are agnostic about when to stop in a time-pressured scenario.

The reviewer expressed concerns about exceeding the page limit, over-citing the C-DAC paper, and some typographical errors. The paper is in compliance with new NIPS guideline of (8+1) pages. We cite Infomax and C-DAC several times, the latter more so owing to our myopic approximation; we will be more parsimonious in a revision. There is a typographical error in Fig. 3 which caused confusion: it should be "Myopic C-DAC" instead of "Greedy MAP". Lastly, a patch is a stimulus location possibly containing the target, a definition we missed.

Reviewer #3:
The reviewer asked for clarification of what "context-sensitivity" means. Based on the visual search task, and the reward structure explained to the subjects (line 236), context-sensitivity refers to the ability to change the control policy based on behavioral costs, such as those related to time, inaccuracy, and effort.

The reviewer asked about the number of free and dependent variables in the policies. C-DAC actually explains all trends in fixation duration (general comment 3), as well as Hits/FAs, and has two new parameters (c & c_s). Since we use a decision threshold for Infomax and Myopic C-DAC that also depends on c, effectively there is just one extra parameter, c_s. The reviewer also made some minor comments regarding formatting, which we intend to incorporate in the revision.